# RATE OF APPROXIMATION BY FLOWS: A CASE STUDY ON THE EIKONAL EQUATION

## ABSTRACT

Previous works have demonstrated the universal approximation capability of residual networks through their continuous idealization as flow maps of dynamical systems. However, informative results on their approximation rates in terms of depth (corresponding to time) are generally lacking. From the viewpoint of approximation theory, a major difficulty in addressing this gap lies in identifying an appropriate target space for the approximation problem. In this paper, we introduce a restrictive but useful target function space comprised of solutions to the eikonal equations, a type of first-order nonlinear partial differential equation, to investigate the approximation rates of flow map families. We provide an estimate of the approximation error within this space, which is notably different from classical rate estimates based directly on the smoothness of target functions. This theoretical result further inspires a new learning-based algorithm for solving the eikonal equation. Experimental results validate the effectiveness of our proposed algorithm, including its robustness to spatial resolution and solution regularity, as well as transferability among similar problems.

## 1 INTRODUCTION

Flow maps of dynamical systems play a crucial role in machine learning, offering both practical applications and theoretical insights. They are particularly valuable in generative modeling due to their inherent dynamical structure and invertibility (Chen et al., 2018; Song et al., 2020; Papamakarios et al., 2021). Furthermore, deep neural networks with residual connections can be idealized as controlled dynamical systems (Weinan, 2017; Haber and Ruthotto, 2017), where the input-output mapping corresponds to a flow map and the network depth relates to the maximal time horizon of the flow. This perspective underscores the significance of understanding the approximation capabilities of flow map families in deep learning theory.

In the context of machine learning, approximation theory studies how well functions in a target space $\mathcal{T}$ can be approximated using a hypothesis space $\mathcal{H}$, which is determined by the choice of model architectures. Two main questions arise in approximation theory: density and approximation rate. The former concerns whether $\mathcal{H}$ is dense in $\mathcal{T}$ in some topology, while the latter involves estimating the approximation error for a function $f \in \mathcal{T}$ in relation to the model's complexity. For example, when $\mathcal{H}$ is the space of polynomials, the Stone-Weierstrass theorem (Stone, 1948) establishes its density in the space of continuous functions. Additionally, Jackson's theorem (Jackson, 1930) provides an approximation rate estimation, indicating that the error in approximating an order-$r$ smooth function $f$ using degree $m$ polynomials is bounded by $C(r, f)m^{-r}$, where $C(r, f)$ is a constant depending on both $r$ and $f$. This approximation rate tells us the number of polynomial terms required to achieve a specific accuracy in approximating $f$. Consequently, approximation rate results are more informative and practical than density results, as they offer a quantitative measure of the cost for approximation, it also tells us which functions can be approximated easily.

Despite established density results for flow map families over a broad range of dynamical systems (Li et al., 2022; Tabuada and Gharesifard, 2022; Ruiz-Balet and Zuazua, 2023; Cheng et al., 2023), the

approximation rate in terms of the maximal time horizon $T$ (analogous to network depth) is not well-understood. Existing results suffer from the curse of dimensionality or are limited to one-dimension case. For instance, Ruiz-Balet and Zuazua (2023) showed that flow maps of continuous-time ResNet

$$\dot{x}(t) = W(t)\sigma(A(t)x(t) + b(t)), \; x(t) \in \mathbb{R}^d, \tag{1}$$

with ReLU activation and piecewise constant parameters chosen from a bounded set can achieve an approximation rate of $\mathcal{O}(T^{-\frac{C}{d^2}})$ in $L^2$ over bounded subsets of $\mathbb{R}^d$, but this rate deteriorates rapidly as the dimension $d$ increases. Similarly, Li et al. (2022) provided rate estimates for $d = 1$, but their results do not generalize to higher dimensions. Consequently, it remains unclear when a deeper flow (larger $T$) offers advantages over a shallower one for a given target function in general. From the perspective of approximation theory, identifying an appropriate target space is crucial in bridging this gap.

In this paper, instead of establishing approximation rate in a general target space, our purpose is to focus on a specific yet practically relevant setting. Specifically, we identify a novel target space comprising solutions to the eikonal equation, a type of first-order nonlinear partial differential equation. By observing that solutions to the eikonal equation can be represented via the flow map of a static vector field, we propose a flow-based hypothesis space and develop an approximation architecture that first approximates the underlying flow map and then computes its flow-based representation to approximate the solution. This architecture resembles a flow idealization of recurrent networks with residual connections, where each layer incorporates specific information about the solution. We establish an approximation rate estimate within this hypothesis space, providing a quantitative measure of the approximation error with respect to the network depth $T$. Unlike classical smoothness-based approximation results DeVore and Lorentz (1993), our estimate depends on the dynamical structures of the solutions, offering a precise setting where increasing depth yields provable performance improvements.

Moreover, our approximation results motivate a novel deep learning method, which we call the *finite flow method*, for solving the eikonal equation. This method numerically implements our proposed flow-based hypothesis space using a deep neural network. The flow-based representation further allows us to train the network by minimizing the variational loss of the represented solution, which is a more efficient way than minimizing the equation loss. Experimental comparisons with the fast marching method Sethian (1999), a state-of-the-art finite difference solver, and existing physics-informed neural network (PINN) methods bin Waheed et al. (2021); Grubas et al. (2023) demonstrate the effectiveness of our algorithm, highlighting advantages in robustness to spatial resolution, solution regularity, and transferability among similar problems.

In summary, the main contributions of this paper are as follows:

- By identifying the solutions to the eikonal equation as a target space, we construct a flow map based hypothesis space and provide an approximation rate estimate in the space (Theorem 2.1). While the target family is restrictive, the resulting approximation rate does not suffer from the curse of dimensionality.

- We show that, different from the classical approximation theory where the approximation error is estimated by the smoothness of the target, our approximation rate result depends on dynamical structures of the solution to the eikonal equation.

- We further show that while the target space is restrictive, it is useful. We develop the *finite flow method* for solving the eikonal equation by numerically implementing our hypothesis space. Our method demonstrates effectiveness compared to state-of-the-art finite difference solvers and existing PINN methods, offering advantages in certain scenarios.

## 2 RELATED WORK

**Approximation rate results of deep neural networks** As a fundamental aspect in deep learning theory, the expressive capability of deep neural networks has been extensively studied. The universal approximation results of different types of deep architectures have been established in the literature (Hornik et al.,

1989; Kidger and Lyons, 2020; Li et al., 2022; Yun et al., 2019). Compared to the universal approximation results, the approximation rates of deep neural networks are less understood. For deep fully connected neural network with ReLU-type activation functions, a series of non-asymptotic approximation rate results are obtained, e.g. Yarotsky (2018); Shen et al. (2019); Lu et al. (2021). Recently, Chenghao et al. extends similar results to ResNet with one-dimension output. These results all suffer from the curse of dimensionality, since their target space are all general smooth function spaces. Some artificial function classes are introduced for curse-of-dimensionlality-free results Montanelli (2021); Poggio et al. (2017); He (2023), but these function spaces lack a clear connection to practical applications. For the family of flow maps, there are fewer results on the approximation rate. By idealizing deep residual networks as the flow map of a continuous-time dynamical system, the authors in Ruiz-Balet and Zuazua (2023) provide an approximation rate of $\mathcal{O}(T^{-\frac{C}{d^2}})$ for deep residual networks with depth $T$ in the target space of all square integrable functions over a bounded set in $\mathbb{R}^d$. This result suffers significantly from the curse of dimensionality, limiting its practical insights. In Li et al. (2022), a nearly optimal rate for the continuous-time ReLU ResNet is obtained in one-dimension case. Although the approach is difficult to extend to higher dimensions, this results reveals a difference of the flow-based approximation to classical smoothness based approximation result.

**Numerical methods for solving the eikonal equation** The eikonal equation, a first-order nonlinear partial differential equation, appears in various applications such as geometric optics, computer vision and seismology Sethian et al. (1999). There are two well-known finite difference methods for the eikonal equation: the fast marching method (FMM) Chopp (2001); Sethian (1999) and the fast sweeping method (FSM) Fomel et al. (2009); Zhao (2005). FMM computes solutions at grid points based on the logic Dijkstra's algorithm, whereas FSM solves the eikonal equation by iteratively sweeping through the grids. Both methods are renowned for their efficiency and accuracy, making them widely adopted in practical applications. The physics-informed neural network (PINN) Raissi et al. (2019) has also been applied to solve the eikonal equation bin Waheed et al. (2021); Grubas et al. (2023); Smith et al. (2020). This approach involves training a neural network to approximate solutions to the eikonal equation by minimizing the equation loss. Several factorization and regularization techniques have been introduced in these works to enhance performance.

## 2.1 TARGET SPACE

For a given $x_s \in \mathbb{R}^d$, let $\mathrm{Lip}(\mathbb{R}^d) \cap C^1(\mathbb{R}^d \setminus \{x_s\})$ be the set of Lipschitz continuous functions in $\mathbb{R}^d$ which is continuously differentiable in $\mathbb{R}^d \setminus \{x_s\}$. For a given point $x_s \in \mathbb{R}^d$, we define the target function space $\Sigma \subset \mathrm{Lip}(\mathbb{R}^d) \cap C^1(\mathbb{R}^d \setminus \{x_s\})$ as:

$$\Sigma := \left\{ u \mid u \geq 0, u(x_s) = 0, |\nabla u| \in \mathrm{Lip}(\mathbb{R}^d \setminus \{x_s\}), 0 < a \leq |\nabla u| \leq b \text{ for some } a, b \in \mathbb{R} \right\}, \quad (2)$$

i.e. there exists Lipschitz continuous function $f_u$ with lower bound $a > 0$ and upper bound $b$, such that $u$ is the *strong* solution of the equation:

$$\begin{cases} |\nabla u(x)| = f_u(x), & x \in \mathbb{R}^n \setminus \{x_s\}, \\ u(x_s) = 0. \end{cases} \quad (3)$$

Equation 3 is called the (one-source) eikonal equation (Tran, 2021). We call $x_s$ the source point and $f_u$ the cost function of the equation. For general Lipschitz cost function, the viscosity solution of the eikonal equation may not be in $\mathrm{Lip}(\mathbb{R}^d) \cap C^1(\mathbb{R}^d \setminus \{x_s\})$ (Tran, 2021). However, that case is not considered in this paper.

## 2.2 HYPOTHESIS SPACE

Let $V_u := -\nabla u/|\nabla u|$ be a continuous unit vector field over $\mathbb{R}^d \setminus \{x_s\}$. By the method of characteristics for eikonal equation, for any $x \in \mathbb{R}^d \setminus \{x_s\}$, it holds that

$$u(x) = \int_0^{\tau(x)} f_u(z(t)) dt, \quad (4)$$

where $z(t)$ satisfies the ODE:

$$\begin{cases} \dot{z}(t) = V_u(z(t)), \\ z(0) = x, \end{cases} \tag{5}$$

$\tau(x)$ is the minimal time of $z(t)$ to the source point $x_s$. See Appendix A.1 for details on this.

Let $\varphi_{V_u}(\cdot, \cdot) : (x, t) \to z(t)$ denote the flow map of equation 23. In the following, we will also use $\varphi_V(\cdot, \cdot)$ to denote the flow of other given vector field $V$ defined in the same way. Replacing $z(t)$ by $\varphi_{V_u}(x, t)$, equation 4 gives a representation of $u$ in terms of the flow map of the static vector field $V_u$. This observation inspires the consideration of approximating $u$ via approximating the corresponding flow map $\varphi_{V_u}$.

To avoid problems caused by the singularity of $V_u$ at the source point, we introduce an auxiliary parameter $\varepsilon > 0$ and define $\chi_\varepsilon$ as the indicator function of the domain $\{x \in \mathbb{R}^d \mid \|x - x_s\| > \varepsilon\}$. For any $x \in \mathbb{R}^d \setminus \{x_s\}$, notice that

$$u(x) = \int_0^{\tau(x)} f_u(\varphi_{V_u}(x, t))dt = \lim_{\varepsilon \to 0} \int_0^T \chi_\varepsilon(\varphi_{\chi_\varepsilon V_u}(x, t)) f_u(\varphi_{\chi_\varepsilon V_u}(x, t))dt, \tag{6}$$

for all $T > \tau(x)$. Therefore, over a bounded subset $\tilde{\Omega}$ of $\mathbb{R}^d \setminus \{x_s\}$, with a small error in representing $u$, we can avoid the trajectories of the flow touching the singularity of $V_u$, and uniformize the traveling time to the origin by an upper bound of $\tau(x)$ over $\tilde{\Omega}$ if we force the trajectory of $\varphi_{V_u}(x, \cdot)$ to stop when it reaches the $\varepsilon$-neighborhood of the source point.

For given $T > 0$, and vector field $V$ in $\mathbb{R}^d$, we define:

$$\Gamma_u^{T,\varepsilon}(V)(x) := \int_0^T \chi_\varepsilon(\varphi_{\chi_\varepsilon V}(x, t)) f_u(\varphi_{\chi_\varepsilon V}(x, t))dt, \tag{7}$$

as long as $\varphi_{\chi_\varepsilon V}$ is well-defined over $[0, T]$, i.e. the solution of the corresponding ODE exists and is unique. We then introduce our hypothesis space to approximate $u \in \Sigma$ in the following definition:

**Definition 2.1.** *Suppose $\mathcal{W}$ is a family of Lipschitz continuous vector field in $\mathbb{R}^d$. For auxiliary parameter $\varepsilon > 0$ and maximal time horizon $T$, we define our hypothesis space to approximate the flow map $u \in \Sigma$ as:*

$$\mathcal{H}_T^u(\mathcal{W}, \varepsilon) := \{\Gamma_u^{T,\varepsilon}(V) \mid V = \frac{W}{|W|} \text{ for some } W \in \mathcal{W}, \text{ s.t. } |W| \neq 0 \text{ in } \mathbb{R}^d\}. \tag{8}$$

In the flow view point, each element in $\mathcal{H}_T^u(\mathcal{W}, \varepsilon)$ can be viewed as a continuous-layer deep neural network with depth $T$. Therefore, there are two components that measures the complexity of this hypothesis space: the size of $\mathcal{W}$ and the time horizon $T$, which corresponding to the layer width and depth of the deep neural network, respectively. Intuitively, over a bounded domain $\tilde{\Omega}$, the approximation error using $\mathcal{H}_T^u(\mathcal{W}, \varepsilon)$ will be small if $\mathcal{W}$ is large and $T$ surpasses the maximal hitting time over $\tilde{\Omega}$, and the auxiliary $\varepsilon$ is small.

Compared to the flow map approximation idealized from practical deep ResNet architectures, there are two differences in the hypothesis space $\mathcal{H}_T^u(\mathcal{W}, \varepsilon)$. First, the flow maps used to represent functions in $\mathcal{H}_T^u(\mathcal{W}, \varepsilon)$ are time-invariant, i.e. the map in each layer are the same. Second, the expression $\Gamma_u^{T,\varepsilon}(V)$ is related to the flow map of $V$ at each time $T$, rather than just the final time $T$. These differences are due to the structure of the eikonal equation. Despite these differences, the essence of approximating $u \in \Sigma$ via the hypothesis space $\mathcal{H}_T^u(\mathcal{W}, \varepsilon)$ is still the approximation by flow maps.

## 2.3 Approximation rate result and its consequences

In this section, we will provide an approximation rate estimate for element $u \in \Sigma$ using the hypothesis space $\mathcal{H}_T^u(\mathcal{W}, \varepsilon)$ defined in the previous section. There are two terms in our estimation. The first term is related to the time horizon $T$, the depth of the network. When $T$ is not large enough for all the initial

state in $B(x_s, 1)$ to reach the source point, there will be an error on the set no matter how accurate the vector field $V_f$ is approximated. This term will be zero when $T$ surpasses the maximal hitting time. The second term is dominated by the approximation error of $|x - x_s| V_f$ in $\mathcal{W}$, and also the regularity of $f$. When $T$ surpasses the maximal hitting time in $B(x_s, 1)$, the approximation error will determined by the expressive capability of the vector field hypothesis space $\mathcal{W}$.

Define $C_u := \dfrac{\sup_x |\nabla u|}{\inf_x |\nabla u|}$. By the definition of $\Sigma$, $C_u \geq 1$ is finite. In Appendix B.1, we show that $\tau(x) \leq C_u |x - x_s|$ for all $x \in \mathbb{R}^d \setminus \{x_s\}$. Therefore, $C_u$ gives a uniform upper bound of $\tau(x)$ over $B(x_s, 1)$. Moreover, define $\tilde{V}_u(x) := -|x - x_s| \dfrac{\nabla u}{|\nabla u|}(x)$ and denote $E_{\mathcal{W}}(\tilde{V}_u) := \inf_{W \in \mathcal{W}} \|W - \tilde{V}_u\|_{L^\infty(B(x_s, C_f))}$ as the approximation error of $\tilde{V}_u$ in $\mathcal{W}$. We then have the following theorem:

**Theorem 2.1.** *Assume that $\varepsilon < 1$ and*

$$(2C_u^2 + C_u)(4E_{\mathcal{W}}(\tilde{V}_u))^{\frac{1}{C_u(L+1)+2}} < \varepsilon \tag{9}$$

*Then, for given $T > 0$, and $u \in \Sigma$, we have:*

$$\inf_{\hat{u} \in \mathcal{H}_T^u(\mathcal{W}, \varepsilon)} \|u - \hat{u}\|_{L^1(B(x_s, 1))} \leq C_1 \max\{C_u - T, 0\}^2 + C_2(4E_{\mathcal{W}}(\tilde{V}_u))^{\frac{1}{C_u(L+1)+2}} + C_3 \varepsilon, \tag{10}$$

*where $L$ is the Lipschitz constant of $\tilde{V}_u$. The constants $C_1$, $C_2$ and $C_3$ are given by $C_1 = 2\pi \|f_u\|_\infty / C_u$, $C_2 = \pi L_{f_u}(6C_u^3 + 3C_u^2)$, and $C_3 = 2\pi \|f_u\|_\infty$, with $L_{f_u}$ being the Lipschitz constant of $f_u = |\nabla u|$.*

*Proof idea.* For given $T \leq C_u$, the domain $B(x_s, 1)$ can be divided into two parts: the set where $\tau(x) \leq T$ and the set where $\tau(x) > T$. When the condition equation 9 is satisfied, there exists $\hat{u} \in \mathcal{H}_T^u(\mathcal{W}, \varepsilon)$ whose corresponding flow can steer all points with $\tau(x) \leq T$ to the $\varepsilon$-neighborhood of $x_s$ within time $T$. Therefore, the error for this part can be bounded by an ODE estimation in terms of the difference between $W/|W|$ and $V_u$. For the other part, besides the error of flow map approximation, there is an additional error caused by the insufficient traveling time, which can be bounded via an estimation on the measure of this set. See the detailed proof in Appendix B.2. □

When $\mathcal{W}$ tends to a universal approximation familiy, the right-hand side of the inequality 10 can actually be arbitrarily small as long as $T \geq C_u$. The estimation equation 10 implies that when $T < C_u$ and $E_{\mathcal{W}}(\tilde{V}_u), \varepsilon$ are small, the approximation error is bounded by a quadratic term of $C_u - T$. In general, $C_u$ is only an upper bound of the maximal hitting time over $B(x_s, 1)$. More precisely, we may expect that the error is approximately quadratic in terms of $\tau_{\max} - T$ with $\tau_{\max}$ being the supremum of $\tau(x)$ over $B(x_s, 1)$. We verify this numerically over 2D examples by calculating the flow map using ODE scheme with small step length (see Appendix C.1 for the expression of these functions). Figure 1 shows the $\log - \log$ plot between $(\tau_{\max} - T)$ and the empirical approximation error over $B(x_s, 1)$ when $E_{\mathcal{W}}(\tilde{V}_u)$, $\varepsilon$ are small. The result verifies the approximately quadratic relation between them.

The approximation rate in Theorem 2.1 differs significantly from classical smoothness-based approximation results. For instance, in polynomial approximation, the error in approximating an order-$r$ smooth target function $F$ using polynomials of degree $\leq n$ is bounded by $Cn^{-r}|F|_{W^{r,p}}$, where $|F|_{W^{r,p}}$ denotes the Sobolev norm of $F$ in $W^{r,p}$. This implies that functions that are easy to approximate using polynomials are those with higher order smoothness and small derivatives in each order. Conversely, in our approximation result, the maximal time $T$ is bounded by the ratio between the maximal and minimal gradient norm of $u$. However, any smoothness assumptions on $u$ or $|\nabla u|$ cannot solely guarantee a bound for $C_u$, highlighting a fundamental difference from smoothness-based approximation.

Moreover, when $T$ exceeds the maximal hitting time, the approximation rate in equation 10 is related to the smoothness of $\tilde{V}_u$ if $\mathcal{W}$ is a smoothness-based approximation family. However, the smoothness of $\tilde{V}_u$ and $u$ can differ greatly. For instance, for any $u \in \Sigma$ of the form $u = \alpha(|x - x_s|)$, where $\alpha : [0, \infty) \to [0, \infty)$ is an increasing function, we have $\tilde{V}_u(x) = x - x_s$, which is infinitely differentiable. Nevertheless, by

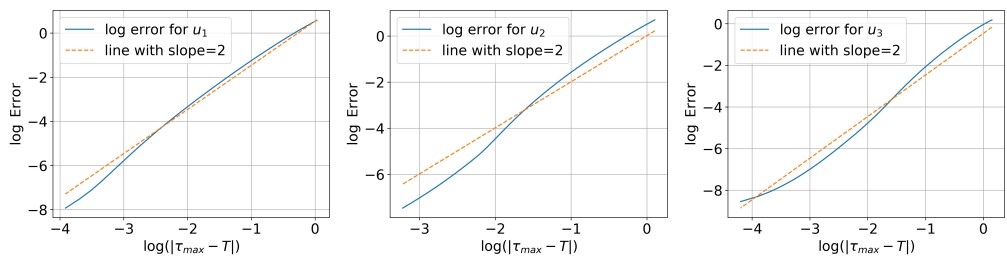

Figure 1: The $\log - \log$ plot between $(\tau_{\max} - T)$ and the empirical approximation error over 3 different cost functions. Lines with slope 2 are plotted for comparison.

the definition of $\Sigma$, such $u$ only needs to be second-order differentiable. Hence, when $\tilde{V}_u$ is sufficiently smooth, the approximation rate to $u$ primarily depends on the smoothness of $\tilde{V}_u$, rather than $u$ itself. This characteristic contrasts with classical approximation schemes where $u(x)$ is approximated by a linear combination of basis functions, in which the smoothness of $u(x)$ controls the approximation quality.

In fact, the example of radial functions is not an isolated case. We provide Proposition B.4 in Appendix B.3, indicating that in general, even if $\tilde{V}_u$ is determined to have high order of smoothness, there still exists a large degree of freedom in $u \in \Sigma$ such that its higher order smoothness cannot be guaranteed.

Let us now also contrast this with currently known rate estimates concerning the depth $T$ of flow maps. In Ruiz-Balet and Zuazua (2023), an estimation of the approximation rate by continuous-time ResNet with activation function ReLU for general $L^2$ integrable target functions is provided. However, their result suffers from the curse of dimensionality, and evaluating the complexity measure of the target function is challenging. Compared to their setting, which approximates general target functions, our approach focuses on a specific class of target functions with dynamical structures. Despite this restrictive setting, we obtain a rate estimate which is curse-of-dimensionality-free with respect to time $T$, indicating when a deeper network should be used over a shallower one and the corresponding error we can expect. Additionally, our identified target functions have the property that their gradient norms are bounded both above and below, with the ratio of these bounds estimating $T$. This setting is analogous to the 1D ReLU flow approximation in Li et al. (2022), where target functions that can be approximated by flow within a finite time horizon have derivatives bounded both above and below. These connections can offer valuable insights into the properties target functions should possess for effective approximation by flow maps.

In summary, our approximation result identifies a restrictive but precise setting where the advantage of using a deeper network can be quantified. Within such a setting, the approximation results by flows are shown to be quite different to the smoothness-based approximation. This phenomenon provides further evidence that smoothness should not be the correct measure of complexity for flow approximation. Moreover, the common feature of our target functions to the previous results may provide valuable insights into the key properties of target functions that can be effectively approximated by flow maps and deep neural networks.

## 3 NUMERICAL IMPLEMENTATION: FINITE FLOW METHOD

In this section, by a numerical implementation of the hypothesis space $\mathcal{H}_T^u(\mathcal{W}, \varepsilon)$, we propose the *Finite Flow Method*, a novel learning-based algorithm to solve the eikonal equation. In this setting of solving PDE, we are assumed to priorly know the right hand side function $f_u$ of equation 3 instead of $u$.

**Representation of the solution** In the finite flow method, we choose the vector field hypothesis space $\mathcal{W}$ to be the set of functions represented by a deep neural network $NN_\theta$ with parameters $\theta$. Subsequently, we calculate the corresponding function $u_\theta$ in $\mathcal{H}_T^u(\mathcal{W}, \varepsilon)$ using an Euler scheme to approximate the real solution $u$. Due to the indicator function introduced to avoid the singularity at the source point, higher

order ODE scheme will essentially be first order near the source point. Therefore, we choose to use the Euler scheme for computational efficiency. Moreover, to avoid differentiability problem to $\theta$ during training, we adopt a smooth approximation of the indicator function $\chi_\varepsilon$ which is adapted to the ODE step size. The details are shown in Appendix C.2.

**Loss Function** According to the variational formulation of the eikonal equation (Tran, 2021), we have

$$u(x) = \inf\{\int_0^T f_u(\varphi_{V_u}(x,t))dt \mid \|V_u\|_\infty \le 1, \text{s.t. } \varphi(x,0) = x, \varphi(x,T) = x_s\}. \tag{11}$$

That is, the integral curve of $V_u$ from $x$ to $x_s$ always minimize the integral of $f_u$ along the curve connecting $x$ and $x_s$. This suggests initializing $u_\theta$ such that its corresponding flow converges to a small neighborhood of $x_s$, as realized in Appendix C.2. We then train the network by minimizing the following loss function to evaluate the solution over the bounded domain $\Omega$:

$$L(\theta) := \|\bar{u}_\theta\|_p^p = \int_\Omega |\bar{u}_\theta(x)|^p dx, \quad p \ge 1. \tag{12}$$

Intuitively, as long as the learning rate is small, the flow of $u_\theta$ will keep converging to a small neighborhood of $x_s$ during training; otherwise, $L(\theta)$ will increase during the training steps. In practice, we can simply evaluate equation 12 by the empirical average over a set of samples. Notice that the loss 12 does not incorporate the derivative of the neural network. Therefore, during the training process, only the first order derivative of the neural network is calculated. This is more computationally efficient than minimizing the equation loss, as used in the physics-informed neural networks(PINN). A diagram of the finite flow method is shown in figure 2. Here, we use finite residual blocks to approximate the flow map representing the solution, and that is why we call it the finite flow method. Suppose we are using the ODE size $\Delta t$ to calculate the flows, and the time horizon is $T = N\Delta t$, then the network for the finite flow method consists of $N$ repeated blocks with residual connections, where the output at each block all contributes to the final output.

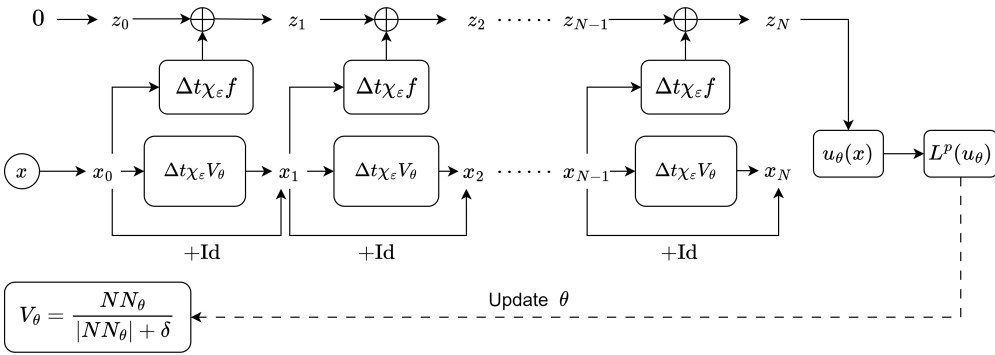

Figure 2: A diagram of the finite flow method. Here $z_n = z_{n-1} + \Delta t \chi_\varepsilon(x_{n-1})f(x_{n-1})$ records a sum over the history outputs at each block. Therefore, the final output is actually related to the state at each block, instead of only the last block in most neural network architectures.

## 4 EXPERIMENTS

In this section, we provide numerical test of our proposed finite flow method in solving the one-source eikonal equation 3. In the experiments, we compare our method to the fast marching method, one of the state-of-the-art finite difference methods for solving the eikonal equation, and existing PINN methods PINNeik (bin Waheed et al., 2021) and NES-OP (Grubas et al., 2023). All the experiments presented in this section solves the one-source eikonal equation 3 over $\Omega = [-1, 1]^2 \subset \mathbb{R}^2$ with $x_s = (0, 0)$. Additional numerical details are provided in Appendix C.3.

### 4.1 COMPARISON TO FAST MARCHING METHOD

#### 4.1.1 ROBUSTNESS TO SPATIAL RESOLUTION OVER DATA

We first demonstrate the robustness of the finite flow method to the spatial resolution comparing to the fast marching method(FMM). To do this, we consider a cost function $f$ corresponding to the solution function:

$$u(x,y) = \sqrt{x^2 + y^2}(0.5(x - 0.5)^2 + \sin(y) + 3) \tag{13}$$

For the finite flow method, we use grid points from various mesh sizes in the domain $[-1,1]^2$ as training data to train the neural network. As a comparison, we also solve the equations on each such grid using the second order FMM. We then evaluate the mean absolute error (MAE) compared to the exact solution for both the finite flow method and FMM. The results are shown in Figure 3a. It can be seen that the error of the fast marching method is almost linear to the spatial resolution in data. In comparison, the performance of the finite flow method is robust to the spatial resolution of the training data. A relatively small number of training data can result in a high accuracy, compared to the FMM. This advantage can be useful in high-dimensional problems where small grid size is in general not available.

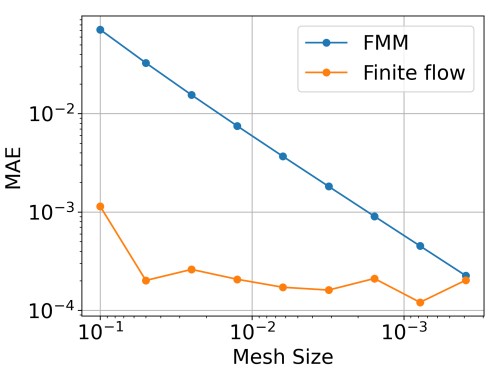 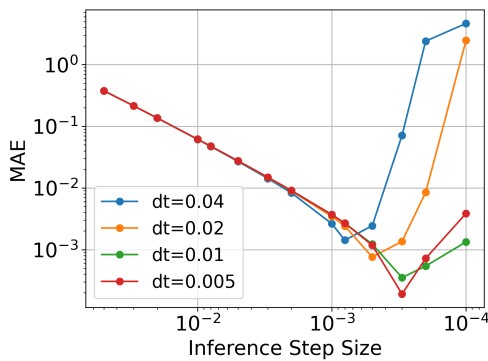

(a) Error vs. Spatial Resolution

(b) Error vs. ODE step size of finite flow method

Figure 3: Factors that affect the accuracy of finite flow method. (a) The mean absolute error of the solution for finite flow method trained over grids with different mesh size. (b) The relation between the solution accuracy and the ODE step size used to inference the solution for networks trained with ODE step size $0.04, 0.02, 0.01$ and $0.005$.

Besides the spatial resolution, there is another more important parameter that affects the accuracy of finite flow method, which is the time step of the ODE scheme. We investigate the effect of different time step length in both the training and evaluation process to the solution accuracy of finite flow method. The results are shown in Figure 3b. It can be seen that we can use a smaller time step length in the evaluation process to achieve a higher accuracy, while the training process can be done with a larger time step length for efficiency. However, when the time step in the evaluation step goes below a certain threshold, the accuracy will start to decrease. In our experiments, we observe that the ratio of the optimal time step length in the evaluation process to the training process should be between $0.02$ to $0.05$.

#### 4.1.2 TRANSFERABILITY AMONG SIMILAR PROBLEMS

In this section, we present the transferability of finite flow method amoung similar problems. Specifically, we consider the scenario where the cost function is perturbed from the original one. We study if the finite flow method trained on the original cost function can adapt to the new cost function within minor computational cost. To do this, we consider train a network for a cost function $f_{\text{original}}(x, y) = 1 + 1.5e^{-3((x-0.4)^2 + (y-0.4)^2)}$ to achieve an MAE of $1.05 \times 10^{-3}$. A perturbation function $f_\delta(x, y) =$

$\delta e^{-3((x+0.4)^2+(y+0.4)^2)}$ with a scale factor $\delta$ is then added to the cost function to get $f_{\text{perturbed}}(x, y) :=$ $f_{\text{original}}(x, y) + f_\delta(x, y)$. We then use the trained network over $f_{\text{original}}$ as an initialization for $f_{\text{perturbed}}$. For a given perturbation scale $\delta$, we compute the number of training steps needed for the pre-trained network to achieve an MAE of $10^{-3}$ for the perturbed cost function. In Figure 4, we plot the ratio of the transfer learning steps to the number of training steps in solving the original equation for $f$. In comparison, we also plot the ratio of the computation time of the FMM in solving the perturbed problem to the original problem. The results show that for finite flow method, the pre-trained network can significantly reduce the computational cost for solving the perturbed cost function, especially when the perturbation is small. In comparison, the computational cost of the fast marching method keeps almost the same as solving the original problem. This result shows the potential of the finite flow method in significantly reducing the computational cost in solving similar problems, particularly in large scale and high-dimensional problems.

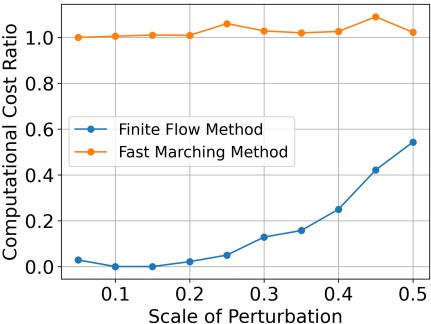

Figure 4: The ratio of additional computational cost for solving the perturbed cost function to the cost for solving the original problem. For finite flow method, the cost is measured by the number of additional training steps needed to achieve the MAE of $10^{-3}$ for the perturbed cost function. For FMM, the cost is measured by the computation time for solving the perturbed equation.

### 4.2 ROBUSTNESS TO SOLUTION REGULARITY COMPARED TO PINN

In this part, we provide a comparison to existing PINN methods for the eikonal equation, NES-OP (Grubas et al., 2023) and PINNeik (bin Waheed et al., 2021).

The key difference of finite flow method to PINN is that finite flow method uses a network to formulate the vector field $V_u = -\nabla u/|\nabla u|$ and represent the solution using the flow-based representation, while the PINN directly use the network output to represent the solution $u$. Intuitively, this implies that PINN is suitable to problems where $u$ is regular, while the finite flow method will perform well when $\nabla u/|\nabla u|$ is regular. In general, when $u$ is smooth enough, $\nabla u/|\nabla u|$ will should also be smooth. However, as we discussed in previous sections, the converse does not hold. Therefore, we may expect that in cases when $u$ and $V_u$ are both regular, both PINN and finite flow method can work well. Moreover, there should also exist cases when $\nabla u/|\nabla u|$ is simple but $u$ itself is not, such that the finite flow method will perform better.

To validate this understanding, we test the performance of the three methods for two cost functions $f_1$ and $f_2$, see Appendix C.3 for the form of $f_1$ and $f_2$. Here, $f_1$ is a simple smooth cost function without sharp spatial change, while $f_2$ is $f_1$ plus a perturbation with relatively small scale but a sharp spatial oscillation. This perturbation does not change the vector field for $f_1$ too much. The results of the three methods are compared in Table 1. We can see that for $f_1$, the best case of all the three methods perform well. PINNeik achieves the highest accuracy in the best case, while finite flow method is more stable in the typical case. In the case of $f_2$, finite flow method outperforms both the PINN methods. Both PINNeik and NES-OP show a significant drop in performance for $f_2$ compared to $f_1$. The result validates that comparing to

the PINN methods, the finite flow method is more robust to the regularity of the solution as long as the regularity of the vector field $V_u$ is still good. This shows a potential of the finite flow method to handle equations with highly heterogeneous cost functions, which is commonly faced in the applications of geophysics (Virieux and Operto, 2009).

| | | Average | Best | Worst |
|---|---|---|---|---|
| **MAE for** $f_1$ | Finite Flow | $\mathbf{1.06 \times 10^{-4}}$ | $9.66 \times 10^{-5}$ | $\mathbf{1.23 \times 10^{-4}}$ |
| | PINNeik | $7.17 \times 10^{-2}$ | $\mathbf{8.02 \times 10^{-5}}$ | $3.58 \times 10^{-1}$ |
| | NES-OP | $5.09 \times 10^{-4}$ | $4.49 \times 10^{-4}$ | $5.16 \times 10^{-4}$ |
| **MAE for** $f_2$ | Finite Flow | $\mathbf{6.05 \times 10^{-4}}$ | $\mathbf{5.92 \times 10^{-4}}$ | $\mathbf{6.32 \times 10^{-4}}$ |
| | PINNeik | $1.44 \times 10^{0}$ | $4.76 \times 10^{-3}$ | $3.54 \times 10^{0}$ |
| | NES-OP | $1.58 \times 10^{-3}$ | $1.32 \times 10^{-3}$ | $1.70 \times 10^{-3}$ |

Table 1: Comparison of the solution accuracy of finite flow method, PINNeik, and NES-OP over $f_1$ and $f_2$. All three methods are trained over the $200 \times 200$ regular grids of $[-1, 1]^2$ using batch size of 10000. The MAE is evaluated over the $101 \times 101$ regular grid points of $[-1, 1]^2$. 5 independent runs are performed for each method, and the best, worst and average MAE are reported.

## 5 LIMITATIONS AND FUTURE WORK

Currently, our theory and proposed numerical solver only handles the strong solution case. In the general case when the characteristic lines of the eikonal equation have intersections, the network in the finite flow method may not learn the correct directions near these intersections. However, according to the variational form of the solution, the flow-based representation is still valid in this case, except that the vector field $V_u$ can have discontinuities over a measure-zero set. In this direction, a direct future work is to design algorithm for weak solution case of the eikonal equation. Moreover, we believe the idea of identifying target space through solutions of PDEs can be generalized to a broader setting, especially for the Hamilton-Jacobi equations (Arnol'd, 2013). In the future work, we will explore the approximation results for other non-linear target spaces derived from non-linear PDEs with inspired learning algorithms for solving the PDE.

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

# A  PROPERTIES OF THE EIKONAL EQUATIONS

## A.1  LINE OF CHARACTERISTICS

Considering solving equation 3 using the method of characteristics(Evans, 2022), the characteristic ODE reads as:

$$
\begin{cases}
\dot{x}(t) = D_p H = \frac{p(t)}{|p(t)|} \\
\dot{p}(t) = -D_x H = -\nabla f(x(t)) \\
\dot{z}(t) = p \cdot D_p H = |p(t)| = f(x)
\end{cases}
\tag{14}
$$

with the initial condition $x(0) = x_s$, $z(0) = 0$ and $p(0) \in \mathbb{R}^d$. Here, $p$ and $z$ stands for the value of $\nabla u$ and $u$ along the characteristics, respectively. Different initial conditions $p(0)$ correspond to different characteristic lines. If the strong solution of equation 3 exist, the trajectory of $x$ in equation 14 will not intersect in $\mathbb{R}^d \setminus \{x_s\}$. Therefore, for any $u \in \Sigma$, considering solving the backward ODE of equation 14 with the initial condition $x(0) = x$ gives the flow-based representation 4 in Section 2.2.

## A.2  VARIATIONAL FORMULATION OF THE EIKONAL EQUATION

The solution to the eikonal equation has a variational formulation.

**Proposition A.1.** *Given any $u \in \Sigma$, for any $x \in \mathbb{R}^d \setminus \{x_s\}$, the following holds:*

$$
\begin{aligned}
u(x) &= \inf \left\{ T : \gamma \in \mathrm{AC}\left([0,t], \mathbb{R}^n\right), \gamma(0) = x, \gamma(T) = x_s, |\gamma'(s)| \leq \frac{1}{f_u(\gamma(T))} \ a.e. \right\} \\
&= \inf \left\{ \int_0^T f_u(\gamma(t)) \mathrm{d}t : \gamma \in \mathrm{AC}\left([0,T], \mathbb{R}^n\right), \gamma(0) = 0, \gamma(T) = x_s, |\gamma'(t)| \leq 1 \ a.e. \right\}.
\end{aligned}
\tag{15}
$$

i.e. the solution $u$ is the value function of the optimal control problem of reaching the source point $x_s$ with the minimal cost under the spatial cost function $f_u$. The locally optimal control is just given by the vector field $V_u = -\nabla u / |\nabla u|$. This property also provides an optimality property of the flow-based representation 4, which is the basis for the loss function used in our proposed finite flow method. We refer to Tran (2021) for the proof of this proposition. In fact, the variational formulation holds in the more general setting of viscosity solutions.

# B  PROPOSITIONS AND PROOFS

## B.1  ESTIMATION ON $\tau(x)$

We have the following proposition on the estimation of $\tau(x)$, which is a direct consequence of the variational formulation of the eikonal equation.

**Proposition B.1.** *For any $u \in \Sigma$, denote*

$$
C_u := \sup_x |\nabla u(x)| / \inf_x |\nabla u(x)| \in [1, \infty).
\tag{16}
$$

*Then, for any $x \in \mathbb{R}^d \setminus \{x_s\}$, we have*

$$
\tau(x) \leq C_u \|x - x_s\|.
\tag{17}
$$

*Proof.* By Proposition 15, we have

$$
\tau(x) \inf_x |\nabla u(x)| \leq u(x) \leq \tau(x) \sup_x |\nabla u(x)|,
\tag{18}
$$

which completes the proof. $\qquad\square$

## B.2 PROOF OF THEOREM 2.1

$$\mathcal{A}_{\mathcal{W}}(\varepsilon) := \{\varphi_{\chi_\varepsilon V} \mid V = \frac{W}{|W|} \quad \text{for some} \quad W \in \mathcal{W}\}, \tag{19}$$

as the flow map family corresponds to the hypothesis space $\mathcal{H}_T^u(\mathcal{W}, \varepsilon)$. We will first provide an approximation error estimate of $\varphi_{V_u}$ over $B(x_s, 1)$ using the hypothesis space $\mathcal{A}_{\mathcal{W}}(\varepsilon)$.

**Theorem B.2.** *Suppose $\varepsilon$. Define $\tilde{V}_u(x) := |x - x_s| V_u(x)$. Then, there exists $\varphi \in \mathcal{A}_{\mathcal{W}}(\varepsilon)$ such that*

$$|\varphi(x, t) - \varphi_{V_u}(x, t)| \le C(4 E_{\mathcal{W}}(\tilde{V}_u))^{\frac{1}{C_u(L+1)+2}} + 2\varepsilon, \tag{20}$$

*for all $x, t$ with $|x - x_s| \le 1$ and $t \le \tau(x)$. Here, $L$ is the Lipschitz constant of $\tilde{V}_u$, $C = 6C_u^2 + 3C_u$, $E_{\mathcal{W}}(\tilde{V}_u) := \inf_{W \in \mathcal{W}} \|W - \tilde{V}_u\|_{L^\infty(B(x_s, C_u))}$ represents the approximation error of $\tilde{V}_u$ in $\mathcal{W}$.*

*Proof.* By the definition of $E_{\mathcal{W}}(\tilde{V}_u)$, there exists $W \in \mathcal{W}$ such that

$$|W(x) - |x - x_s| V_u(x)| \le 2 E_{\mathcal{W}}(\tilde{V}_u) \tag{21}$$

for all $x$ s.t. $|x - x_s| \le C_u$. Let $x(t)$ denote the solution of the ODE:

$$\begin{cases} \dot{x}(t) = V_u(x(t)), \\ x(0) = x. \end{cases} \tag{22}$$

Let $\tilde{x}(t)$ denote the solution of the ODE:

$$\begin{cases} \dot{\tilde{x}}(t) = \frac{W(\tilde{x})}{|W(\tilde{x})|}, \\ \tilde{x}(0) = x. \end{cases} \tag{23}$$

For $t \in [0, \tau(x)]$, we have

$$\frac{d}{dt}|x(t) - \tilde{x}(t)| \le \left| V_u(x(t)) - \frac{W(\tilde{x})}{|W(\tilde{x})|} \right| \le |V_u(x(t)) - V_u(\tilde{x}(t))| + \left| V_u(\tilde{x}(t)) - \frac{W(\tilde{x})}{|W(\tilde{x})|} \right|. \tag{24}$$

For the first term, from $||x| V_u(x) - |\tilde{x}| V_u(\tilde{x})| \le L|x - \tilde{x}|$, we have

$$|x - x_s| |V(x) - V(\tilde{x})| \le L|x - \tilde{x}| + |(|x - x_s| - |\tilde{x} - x_s|)| V(\tilde{x})| \le (L+1)(|x - \tilde{x}|), \tag{25}$$

i.e.

$$|V(x) - V(\tilde{x})| \le (L+1) \frac{|x - \tilde{x}|}{|x - x_s|}. \tag{26}$$

For the second term, we have

$$\left| V(\tilde{x}(t)) - \frac{W(\tilde{x}(t))}{|W(\tilde{x}(t))|} \right| \le \frac{1}{|W(\tilde{x})|} \left( |(|W(\tilde{x})| - |x - x_s|) V(\tilde{x})| + |W(\tilde{x}) - |\tilde{x} - x_s| V(\tilde{x})| \right)$$
$$\le \frac{4 E_{\mathcal{W}}(\tilde{V}_u)}{|W(\tilde{x})|} \tag{27}$$

Therefore, we have

$$\frac{d}{dt}|x(t) - \tilde{x}(t)| \le (L+1) \frac{|x - \tilde{x}|}{|x - x_s|} + \frac{4 E_{\mathcal{W}}(\tilde{V}_u)}{|W(\tilde{x})|}. \tag{28}$$

From Proposition B.1, $|x(t) - x_s| \ge \frac{1}{C_u}(\tau - t)$. We have

$$|W(\tilde{x}(t))| + \ge |\tilde{x}(t) - x_s| - E_{\mathcal{W}}(\tilde{V}_u) + \ge \frac{1}{C_u}(\tau - t) - |x(t) - \tilde{x}(t)| - E_{\mathcal{W}}(\tilde{V}_u). \tag{29}$$

We then have

$$\frac{d}{dt}|x(t) - \tilde{x}(t)| \leq \frac{C_u(L+1)}{\tau - t}|x - \tilde{x}| + \frac{4E_{\mathcal{W}}(\tilde{V}_u)}{\frac{1}{C_u}(\tau - t) - |x(t) - \tilde{x}(t)| - E_{\mathcal{W}}(\tilde{V}_u)}, \tag{30}$$

as long as the right-hand side of equation 29 is positive. It follows that

$$|x(t) - \tilde{x}(t)| \leq (\tau - t)^{-C_u(L+1)} \int_0^t \frac{(4E_{\mathcal{W}}(\tilde{V}_u))(\tau - s)^{C_u(L+1)}}{\frac{1}{C_u}(\tau - s) - |x(s) - \tilde{x}(s)| - E_{\mathcal{W}}(\tilde{V}_u)}ds. \tag{31}$$

The following lemma holds:

**Lemma B.3.** *Suppose* $\tau > 2C_u^2(4E_{\mathcal{W}}(\tilde{V}_u))^{\frac{1}{C_u(L+1)+2}}$ *and* $4E_{\mathcal{W}}(\tilde{V}_u) \leq \frac{1}{2C_u}$. *When* $t \leq \tau - 2C_u^2(4E_{\mathcal{W}}(\tilde{V}_u))^{\frac{1}{C_u(L+1)+2}}$, *we have*

$$|x(t) - \tilde{x}(t)| \leq C_u(4E_{\mathcal{W}}(\tilde{V}_u))^{\frac{1}{C_u(L+1)+2}}. \tag{32}$$

*Proof of the Lemma.* Prove by contradiction. Denote $C_u(4E_{\mathcal{W}}(\tilde{V}_u))^{\frac{1}{C_u(L+1)+2}}$ as $\alpha$. Suppose the conclusion does not hold, take $t_0$ as the infimum of the $t$ in $[0, \tau - \alpha]$ such that the inequality fails. Since $|x(0) - \tilde{x}(0)| = 0$ and $|x(t) - \tilde{x}(t)|$ is continuous, we have $t_0 > 0$. From equation 31, we have

$$|x(t) - \tilde{x}(t)| \leq (\tau - s)^{-C_u(L+1)} \int_0^t \frac{(4E_{\mathcal{W}}(\tilde{V}_u))(\tau - t)^{C_u(L+1)}}{\frac{1}{C_u}(\tau - s) - \alpha - E_{\mathcal{W}}(\tilde{V}_u)}ds$$

$$\leq (\tau - t)^{-C_u(L+1)} \int_0^t \frac{(4E_{\mathcal{W}}(\tilde{V}_u))\tau^{C_u(L+1)}}{\frac{1}{C_u}(\tau - s) - \alpha - E_{\mathcal{W}}(\tilde{V}_u)}ds$$

$$= \frac{(4E_{\mathcal{W}}(\tilde{V}_u))\tau^{C_u(L+1)}}{(\tau - t)^{C_u(L+1)}}C_u \log(\frac{1}{C_u}(\tau - t) - \alpha - E_{\mathcal{W}}(\tilde{V}_u))\Big|_0^t \tag{33}$$

$$\leq \frac{C_u(4E_{\mathcal{W}}(\tilde{V}_u))\tau^{C_u(L+1)}}{(\tau - t)^{C_u(L+1)}} \log \frac{\tau}{\tau - t - C_u(\alpha + E_{\mathcal{W}}(\tilde{V}_u))}$$

$$\leq \frac{C_u(4E_{\mathcal{W}}(\tilde{V}_u))\tau^{C_u(L+1)}}{(\tau - t)^{C_u(L+1)}}(\frac{\tau}{\tau - t - C_u(\alpha + E_{\mathcal{W}}(\tilde{V}_u))} + 1)$$

for $[0, t_0)$. By continuity, we have

$$|x(t_0) - \tilde{x}(t_0)| \leq \frac{C_u(4E_{\mathcal{W}}(\tilde{V}_u))\tau^{C_u(L+1)}}{(\tau - t_0)^{C_u(L+1)}}(\frac{\tau}{\tau - t_0 - C_u(\alpha + E_{\mathcal{W}}(\tilde{V}_u))} + 1)$$

$$\leq \frac{C_u(4E_{\mathcal{W}}(\tilde{V}_u))\tau^{C_u(L+1)}}{(2C_u\alpha)^{C_u(L+1)}}(\frac{\tau}{2C_u\alpha - C_u(\alpha + E_{\mathcal{W}}(\tilde{V}_u))} + 1) \tag{34}$$

$$\leq \frac{(4E_{\mathcal{W}}(\tilde{V}_u))^{\frac{2}{C_u(L+1)+2}}}{(2C_u^2)^{C_u(L+1)}}\frac{2(\tau)}{\alpha}$$

$$\leq (4E_{\mathcal{W}}(\tilde{V}_u))^{\frac{1}{C_u(L+1)+2}} \leq \alpha,$$

which is a contradiction. Here we use $C_u \geq 1$, $\tau \leq C_u$. $\qquad\square$

The Lemma gives an estimate of $|x(t) - \tilde{x}(t)|$ when $t \leq \tau - 2C_u^2(4E_{\mathcal{W}}(\tilde{V}_u))^{\frac{1}{C_u(L+1)+2}}$. For $t \in [\tau - 2C_u^2(4E_{\mathcal{W}}(\tilde{V}_u))^{\frac{1}{C_u(L+1)+2}}, \tau]$ and the case when $\tau \leq 2C_u^2(4E_{\mathcal{W}}(\tilde{V}_u))^{\frac{1}{C_u(L+1)+2}}$, we can consider the simplest estimation that

$$\frac{d}{dt}|x(t) - \tilde{x}(t)| \leq 2 \tag{35}$$

Combine with the result of the Lemma, we have:

$$|x(t) - \tilde{x}(t)| \le (2C_u^2 + C_u)(4E_\mathcal{W}(\tilde{V}_u))^{\frac{1}{C_u(L+1)+2}}, \tag{36}$$

for all $t \in [0, \tau(x)]$. Now denote $\tilde{V} := \frac{W}{|W|}$. We have actually shown that

$$|\varphi_{\tilde{V}}(x,t) - \varphi_{V_u}(x,t)| \le (2C_u^2 + C_u)(4E_\mathcal{W}(\tilde{V}_u))^{\frac{1}{C_u(L+1)+2}}, \tag{37}$$

for all $|x - x_s| \le 1$ and $t \in [0, \tau(x)]$. Notice that the trajectory of $\tilde{V}_u$ and $\chi_\varepsilon V_u$ with the same initial condition $x$ will be the same before the first time $t_\varepsilon$ when $|\tilde{x}(t) - x_s| = \varepsilon$. Since

$$|x(t_\varepsilon) - x_s| \le (2C_u^2 + C_u)(4E_\mathcal{W}(\tilde{V}_u))^{\frac{1}{C_u(L+1)+2}} + \varepsilon, \tag{38}$$

we have

$$t_\varepsilon \ge \tau - (2C_u^2 + C_u)(4E_\mathcal{W}(\tilde{V}_u))^{\frac{1}{C_u(L+1)+2}} - \varepsilon. \tag{39}$$

Therefore, we have

$$|\varphi_{\tilde{V}}(x,t) - \varphi_{V_{\chi_\varepsilon \tilde{V}}}(x,t)| \le 2(\tau - t_\varepsilon) \le 2(2C_u^2 + C_u)(4E_\mathcal{W}(\tilde{V}_u))^{\frac{1}{C_u(L+1)+2}} + 2\varepsilon. \tag{40}$$

It then follows that

$$|\varphi_{V_{\chi_\varepsilon \tilde{V}}}(x,t) - \varphi_{V_u}(x,t)| \le (\tau - t_\varepsilon) \le (6C_u^2 + 3C_u)(4E_\mathcal{W}(\tilde{V}_u))^{\frac{1}{C_u(L+1)+2}} + 2\varepsilon, \tag{41}$$

for all $|x - x_s| \le 1$ and $t \in [0, \tau(x)]$. Since $\varphi_{V_{\chi_\varepsilon \tilde{V}}} \in \mathcal{A}_\mathcal{W}(\varepsilon)$, the theorem is proved. $\qquad \square$

Now we can give the proof of Theorem 2.1

*Proof of Theorem 2.1.* From Theorem B.2 and equation 36 in its proof, we know that when

$$(2C_u^2 + C_u)(4E_\mathcal{W}(\tilde{V}_u))^{\frac{1}{C_u(L+1)+2}} < \varepsilon, \tag{42}$$

there exists $\varphi = \varphi_{\chi_\varepsilon V} \in \mathcal{A}_\mathcal{W}(\varepsilon)$ such that equation 20 holds, and its trajectory will reach the $\varepsilon$-neighborhood of $x_s$ before time $\tau(x)$. That is, $|\varphi(x,t) - x_s| = \varepsilon$ for all $x \in B(0,1)$ and $t > \tau(x)$.

For this $V$ with $\varphi = \varphi_{\chi_\varepsilon V}$ we have

$$\|u - \Gamma_u^{T,\varepsilon}(V)(x)\|_{L^1(B(0,1))} = \int_{B(0,1)} \|u(x) - \hat{u}(x)\| \mathrm{d}x$$

$$= \int_{x \in B(0,1), \tau(x) \le T} \|u(x) - \Gamma_u^{T,\varepsilon}(V)(x)\| \mathrm{d}x + \int_{x \in B(0,1), \tau(x) > T} \|u(x) - \Gamma_u^{T,\varepsilon}(V)(x)\| \mathrm{d}x \tag{43}$$

For the first term, we have

$$\int_{x \in B(0,1), \tau(x) \le T} \|u(x) - \Gamma_u^{T,\varepsilon}(V)(x)\| \mathrm{d}x$$

$$= \int_{x \in B(0,1), \tau(x) \le T} \| \int_0^{\tau(x)} (f_u(\varphi_{V_u}(x,t)) - \chi_\varepsilon(\varphi(x,t)) f_u(\varphi(x,t)dt)) \| \mathrm{d}x$$

$$\le \int_{x \in B(0,1), \tau(x) \le T} \left( \int_0^{\tau(x)} (f_u(\varphi_{V_u}(x,t)) - f_u(\varphi(x,t)dt)) \, \mathrm{d}t + \int_{\max\{\tau(x)-2\varepsilon, 0\}}^{\tau(x)} \|f_u(\varphi(x,t)dt)\| \mathrm{d}t \right) \mathrm{d}x$$

$$\le \int_{x \in B(0,1), \tau(x) \le T} \int_0^{\tau(x)} L_{f_u} \|\varphi_{V_u}(x,t) - \varphi(x,t)\| \mathrm{d}t \mathrm{d}x + 2m(\{x \in B(0,1), \tau(x) \le T\}) \|f_u\|_\infty \varepsilon$$

$$\tag{44}$$

Here we use the estimation in equation 39 on the first hitting time of $\varphi(x, \cdot)$ to the $\varepsilon$-neighborhood of $x_s$. For the second term, we have

$$\int_{x \in B(0,1), \tau(x)>T} \|u(x) - \Gamma_u^{T,\varepsilon}(\varphi)(x)\| \mathrm{d}x$$

$$= \int_{x \in B(0,1), \tau(x)>T} \left\| \int_0^{\tau(x)} \left( f_u(\varphi_{V_u}(x,t)) - f_u(\varphi(x,t)) \mathrm{d}t + \int_T^{\tau(x)} f_u(\varphi(x,t)) \mathrm{d}t \right) \right\| \mathrm{d}x$$

$$\leq \int_{x \in B(0,1), \tau(x)>T} \int_0^{\tau(x)} L_{f_u} \|\varphi_{V_u}(x,t) - \varphi(x,t)\| \mathrm{d}t \mathrm{d}x + ((C_u - T) + 2\varepsilon) m(\{x \in B(0,1), \tau(x)>T\}) \|f_u\|_\infty$$

(45)

Combine both, by equation 20, we have

$$\|u - \Gamma_u^{T,\varepsilon}(V)(x)\|_{L^1(B(0,1))}$$

$$\leq \pi L_u C_u (6 C_u^2 + 3 C_u)(4 E_{\mathcal{W}}(\tilde{V}_u) +)^{\frac{1}{C_u(L+1)+2}} + 2\pi \|f_u\|_\infty \varepsilon + (C_u - T) m(\{x \in B(0,1), \tau(x)>T\}) \|f_u\|_\infty$$

(46)

When $T \leq C_u$, we have

$$m(\{x \in B(x_s, 1), \tau(x) > T\}) = \pi - m(\{x \in B(x_s, 1), \tau(x) \leq T\})$$

$$\leq \pi - m(\{x \in B(x_s, 1), C_u |x - x_s| \leq T\}) = \pi(1 - (\frac{T}{C_u})^2).$$

(47)

When $T > C_u$, it is clear that $m(\{x \in B(x_s, 1), \tau(x) > T\}) = 0$. Combine this with the previous estimation, we have the conclusion of the theorem. □

## B.3 PARTIAL DETERMINATION RELATION BETWEEN $\tilde{V}_u$ AND $u$

**Proposition B.4.** *For $d \geq 3$, suppose $u \in \Sigma$ is $C^2$ in $\mathbb{R}^d \setminus \{x_s\}$. Define*

$$\mathcal{R}_u := \{v \in \Sigma \mid V_v = V_u\}. \tag{48}$$

*Then, for any continuous function $h$ over $\mathbb{R}^d$ such that $0 < a \leq h \leq b$ for some constant $a, b$, and $\nabla h(x) \parallel \nabla u(x)$ for all $x \in \mathbb{R}^d \setminus \{x_s\}$, there exists $v \in \mathcal{R}_u$ such that $|\nabla v| = h|\nabla u|$.*

*Proof.* Since $u$ is $C^2$ in $\mathbb{R}^d \setminus \{x_s\}$, we have that $V_u$ is $C^1$ in $\mathbb{R}^d \setminus \{x_s\}$. We then have

$$h|\nabla u| V_u = -h \cdot |\nabla u| \frac{\nabla u}{|\nabla u|} = -h \nabla u. \tag{49}$$

We then claim that $-h \nabla u$ is the gradient of some $C^1$ function over $\mathbb{R}^d \setminus \{x_s\}$. By Poincaré's lemma, since $\mathbb{R}^d \setminus \{x_s\}$ is simply connected, we only need to check that

$$\partial_j(h \partial_i u) - \partial_i(h \partial_j u) = \partial_j h \partial_i u - \partial_i h \partial_j u = 0, \quad \forall i, j = 1, \cdots, d, \tag{50}$$

which follows directly from the assumption that $\nabla h \parallel \nabla u$. Therefore, we can assume that there exists $w \in C^1(\mathbb{R}^d \setminus \{x_s\})$, such that

$$h|\nabla u| V_u = \nabla w. \tag{51}$$

We can assume that $w(x_s) = 0$. Then, we can deduce that $-w$ is positive in $\mathbb{R}^d \setminus \{x_s\}$. Since $V_u$ is unit, this implies that $|\nabla w| = h|\nabla u|$. According to $u \in \Sigma$ and $0 < a \leq |\nabla h| \leq b$, we deduce that $-w \in \Sigma$. Let $v = -w$, then we have

$$V_v = -\nabla v / |\nabla v| = V_u, \tag{52}$$

indicating that $v \in \mathcal{R}_u$.

□

## C  EXPERIMENTAL DETAILS

### C.1  VERIFICATION OF APPROXIMATION RATE

We numerically verify the approximation rate with $x_s = (0,0)$ and the following 3 functions over $B(x_s, 1) \subset \mathbb{R}^2$.

- $u_1(x, y) = \sqrt{x^2 + y^2}(\sin(x + \frac{1}{2}) + \cos(\frac{y}{2} + 1))$
- $u_2(x, y) = \sqrt{x^2 + y^2}(\sin(x + 2y) + 3)$
- $u_3(x, y) = \sqrt{x^2 + y^2}(e^{-5(x-0.35)^2 - 5(x-0.35)^2} + e^{-5(x+0.35)^2 - 5(x+0.35)^2})$.

The graph of $u_1$, $u_2$, $u_2$ and their corresponding cost functions $f_1$, $f_2$, $f_3$ are shown.

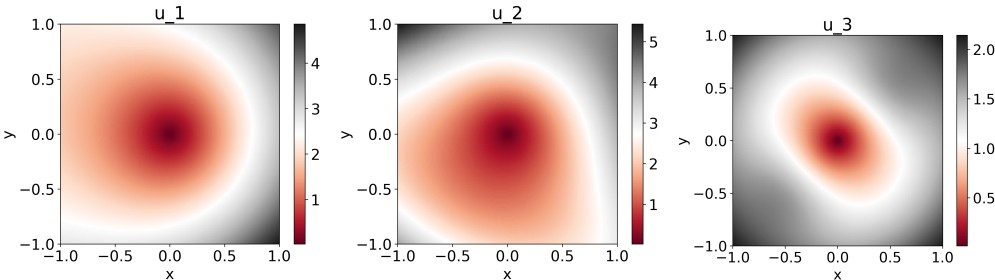

Figure 5: Graph of $u_1, u_2, u_3$

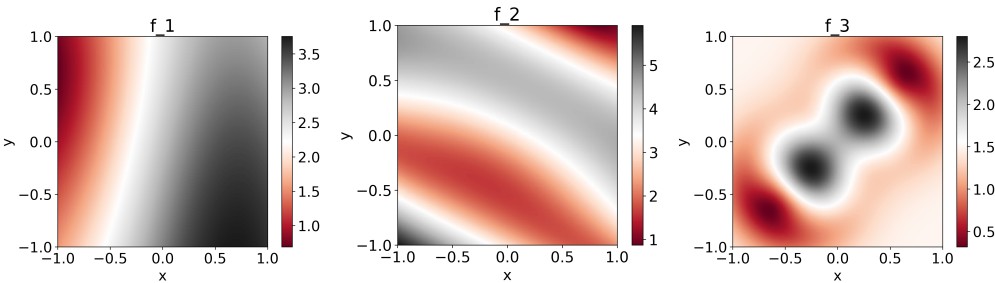

Figure 6: Graph of $f_1, f_2, f_3$

To test approximation error with respect to $T$, for each $u_i (i = 1, 2, 3)$, we simply take $\tilde{V}_{u_i}$ to belong to $\mathcal{W}$ to make $E_{\mathcal{W}}(\tilde{V}) = 0$. Then, for the vector field $V_{u_i}$ and 300 dirrerent values of $T \in [0, 3]$, we use the Euler scheme with step size $10^{-4}$ and $\varepsilon = 3 \times 10^{-4}$ to numerically compute $\Gamma_u^{T,\varepsilon}(V_{u_i})(x)$ over $10^5$ uniformly sampled points in $B(x_s, 1)$ to evaluate the $L^1$ approximation error.

### C.2  DETAILS OF THE FINITE FLOW METHOD

#### C.2.1  NETWORK ARCHITECTURE

Since the underlying vector field $V_u$ of the one-source problem has an attractor at the source point $x_s$, we design the following network architecture to formulate $V_\theta$ in the finite flow method:

$$V_\theta(x) = -\frac{x - x_s + \mathrm{MLP}(x)}{|x - x_s + \mathrm{MLP}(x)|} \tag{53}$$

where MLP is an multilayer perceptron with input and output dimension being the state dimension of the equation. The weight in the output layer of MLP is initialized as zero to ensure that the integral curves of $V_\theta$ is initialized to converge at the source point. This will overcome the issues related to singularity at the source point and make the training dynamics using the variational loss more stable.

### C.2.2 MOLLIFIED INDICATOR FUNCTION

To overcome the differentiability issue caused by the indicator function $\chi_\varepsilon$, we choose a smooth mollifier which is adapted to the ODE stepsize as its substitution. Specifically, in numerical experiments, we replace $\chi_\varepsilon$ with a mollifier $\rho$ which is defined to be $\rho(x) = h(\|x - x_s\|/\Delta t)$, where $\Delta t$ is the stepsize for the Euler scheme, and $h(r)$ is given by $h(r) = \text{softplus}(4(r - 3)) - \text{softplus}(4(r - 3) - 1)$. Such a mollifier can be regarded as a smooth approximation of $\chi_\varepsilon$ for $\varepsilon = 4\Delta t$. The graph of $\rho$ when $\Delta t = 0.01$ is shown below.

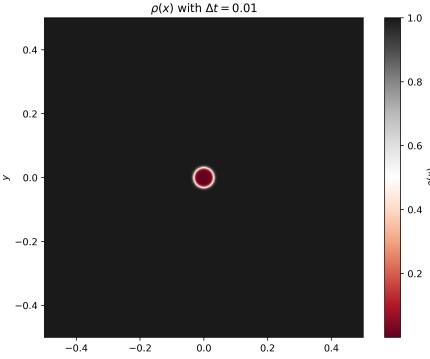

Figure 7: Graph of $\rho$ when $\Delta t = 0.01$

Here, for the numerical stability, we mollify a neighborhood with radius $4\Delta t$ of $x_s$ to ensure that the trajectory calculated using ODE scheme with step length $\Delta t$ will not go out of it once entered.

### C.3 EXPERIMENTAL DETAILS IN SECTION 4

In this section, we provide the implementation details of the experiments in Section 4. For all the experiments on the finite flow method, the network structure follows equation 53 with MLP is a fully connected neural network with 4 hidden layers and 55 neurons in each layer with 9517 trainable parameters. We implement all the learning-based solver using the `equinox` library, which is based on the `JAX` framework. All the training processes use the Adam optimizer. The experiments are conducted on a single NVIDIA RTX 3090 GPU. We perform the second order fast marching method using the `eikonalfm` package.

### C.3.1 IMPLEMENTATION DETAILS IN SECTION 4.1.1

In the first experiment in Section 4.1.1, we compare the finite flow method with the fast marching method on grids with different mesh size. For each given mesh size, we train the network using an ODE step size of $5 \times 10^{-3}$ and a learning rate of $10^{-3}$ with the Adam optimizer. The ODE step in for inference is set to be $2.5 \times 10^{-4}$. The maximal batch size is set to $50000$.

For a fair comparison of the finite flow method and the fast marching method, we evaluate the MAE of finite flow method over the $100 \times 100$ regular grids of $[-1, 1] \times [-1, 1]$, which are mostly not incorporated

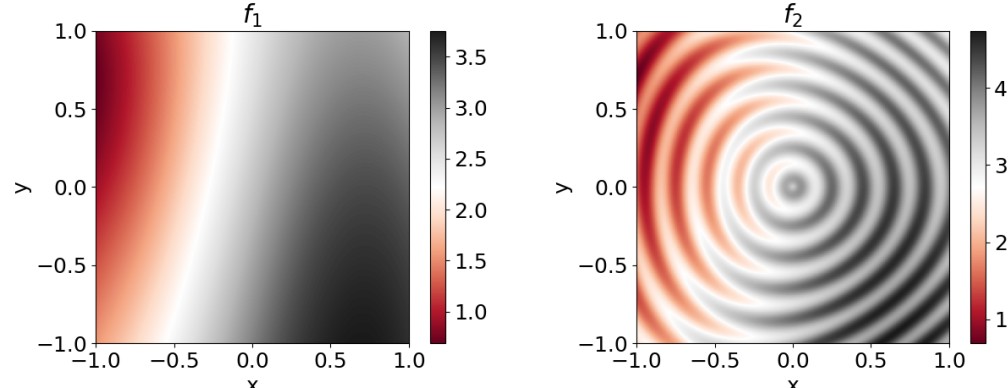

Figure 8: Images of the cost function $f_1$ and $f_2$.

in the training set. The error of the fast marching method is evaluated over the same grids used for solving the equation. This difference in the evaluation set is due to the different nature of the two methods.

In the second experiment in Section 4.1.2, for all the networks, the training set is the $100 \times 100$ regular grids of $[-1, 1] \times [-1, 1]$. A full-batch training with learning rate of $5 \times 10^{-4}$ of are applied to all the networks for 500 epochs.

### C.3.2 Implementation details in Section 4.1.2

In Section 4.1.2, since the explicit solution of the eikonal equation is not available, we use the solution got by the second order fast marching method with mesh size $4 \times 10^{-4}$ as reference.

The network for solving $f_{\text{original}}$ is trained over the $200 \times 200$ regular grids of $[-1, 1] \times [-1, 1]$.

### C.3.3 Implementation details in Section 4.2

The function $f_1$ and $f_2$ used in the experiment are given by

$$f_1(x, y) = \|\nabla u(x, y)\|\|, \text{ with } u = \sqrt{x^2 + y^2}(\sin(x + \frac{1}{2}) + \cos(\frac{y}{2} + 1)),$$

$$f_2(x, y) = f_1(x, y) + \sin(10\pi\sqrt{x^2 + (1 + 0.2x)^2 y^2}) \tag{54}$$

The image of $f_1$ and $f_2$ are shown in Figure 8. The comparison of their corresponding vector fields are shown in Figure 9.

For $f_2$, since the explicit solution is not know, we use the solution got by the second order fast marching method with mesh size $2 \times 10^{-4}$ as reference. For all three networks, the training set is the $200 \times 200$ regular grids of $[-1, 1] \times [-1, 1]$. The batch size is set to be 10000. The networks are all trained with Adam optimizer. Other details of the training setting are listed below:

The network size for the PINNeik and NES-OP is 5 hidden layers with 50 neurons in each layer with 10401 parameters in total, which is close to the network size of the finite flow method.

For the finite flow method, the training epoch is 150 for both $f_1$ and $f_2$ with learning rate $5 \times 10^{-4}$. For the two PINN methods, the training epoch is 30000 for $f_1$ and 60000 for $f_2$ with learning rate $2.5 \times 10^{-4}$.

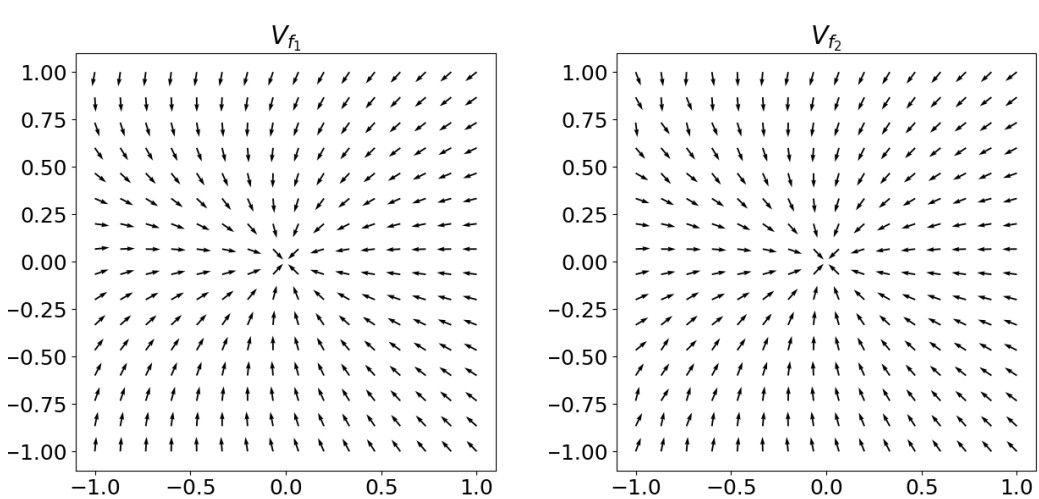

Figure 9: Image of the vector fields $V_1$ and $V_2$ corresponding to $f_1$ and $f_2$, respectively.