# OpenReview forum: "Rate of Approximation by Flows: A Case Study on the Eikonal Equation"
_ICLR.cc/2025/Conference — ICLR 2025 Conference Withdrawn Submission_

### Official Review · Reviewer_ey9J · 2024-11-04

**Soundness:** 2
**Presentation:** 3
**Contribution:** 2
**Rating:** 3
**Confidence:** 3

**Summary:**

The paper proposes the eikonal PDE as a putative model of deep model dynamics, comparing and contrasting to smoothness-based approximation theory.   It also presents work on using deep learning as a numerical solution technique for the eikonal PDE, comparing to the classical fast marching method and to PINNs.

**Strengths:**

The ongoing effort to use PDE methodology to help understand neural network dynamics, and in particular to help reveal when and how depth is needed, is important.  The current paper fits well within the overall effort.  It is clearly presented, and it does a good job of citing related work.

**Weaknesses:**

The paper is two essentially disjoint contributions.  The first is to propose the eikonal equation as a model of deep network dynamics, and the second is to consider deep learning as a solution technique for this class of PDEs.  Each contribution is thus allocated one half of the space, and this compression means that neither idea is explored very deeply.

Unquestionably, the eikonal equation is a PDE of vast importance, in optics and far beyond.  Its connection to optimal control and to Hamilton-Jacobi theory suggests, but only very roughly, a possible connection to neural networks (ResNets), in particular as a way to study the effect of depth.  The authors do make clear (in Section 2) that this is not an entirely compelling connection (in the paragraph starting on line 189).  The problem, though, is that one is left without much conviction about the eikonal equation as a model.  The rest of the section is devoted to criticizing (rightly) smoothness-based approximations, but the criticism doesn't imply that the eikonal equation is the right way to proceed.  Further work is needed on this.

In section 3, the paper turns to a rather different class of problem, which is the numerical solution of the eikonal PDE via deep learning.  This material is better developed, but again one is left with a feeling that more work is needed.  The numerical solution of the eikonal equation is the subject of a vast literature, and the putative advantages of neural network solutions needs a more sustained investigation.  Also, the restriction to strong solutions is quite restrictive (as the authors are clear in pointing out).

**Questions:**

My main question is whether the authors have considered writing two separate papers on the two contributions here, or whether they feel that there's an important connection between the two halves that I have overlooked in my reading.

---

### Official Review · Reviewer_uMHZ · 2024-11-04

**Soundness:** 3
**Presentation:** 2
**Contribution:** 3
**Rating:** 8
**Confidence:** 1

**Summary:**

This paper introduces a function space comprised of solutions to the eikonal equations to investigate the approximation rates of flow map
families. A flow map based hypothesis space is constructed and an approximation rate estimate in the space is provided. This rate result does not suffer from a curse of dimensionality like the results in the previous literature and depends on dynamical structures of the solution to the eikonal equation.

Moreover, a new method for solving the eikonal equation by numerically implementing this hypothesis space is introduced and named "finite flow".

**Strengths:**

The scope of this paper is clear, its contributions are clearly identified and properly contextualized taking into account the previous literature. The reviewer is not familiar with this literature, but the results seem to be novel.

I was not able to review all proofs in great detail.

**Weaknesses:**

I believe that the exposition of the intuition behind the main results needs to be improved. In particular, I found the discussion after Thm. 2.1 to be hard to follow.


Also, some polishing is needed. I have found several typos and sentences with references to equations that do not flow correctly.

**Questions:**

- Do the authors mean to refer to eq. (23) after eq. (5)?

 - Do the authors have any intuition on why the accuracy decreases for certain step-sizes in Figure 3.b? Might there be a bias proportional to the step-size in this case.

---

### Official Review · Reviewer_chqm · 2024-11-08

**Soundness:** 3
**Presentation:** 2
**Contribution:** 2
**Rating:** 5
**Confidence:** 3

**Summary:**

The approximation of neural nets is a well-positioned topic in ML.

This paper does not focus on the typical nets but studies the approximation rates of *flow maps* based on residual networks. It uses a specific target space of solutions to the so-called *Eikonal equation* which is a special case of the Hamilton-Jacobi Equation, offering a new approach to estimating approximation error.
It also introduces an algorithm for solving the eikonal equation. Experiments show the algorithm's robustness across different spatial resolutions for the eikonal equation.

**Strengths:**

- the paper mitigates the issue of previous net approximation results for flow maps, suffering from the curse of dimensionality. In other words, (1) one previous work provides a dimension-dependent approximation rate, and (2) another one only focuses on the one-dimensional case.
- provides a new method for solving the eikonal equation

**Weaknesses:**

- A pressing concern is the unreadability of the paper to an ML audience. I have some background in dynamical systems and ODEs but still found it hard to understand the ideas and the contributions.  I list some examples below, that I hope help the authors. This is not to say that the paper is not suitable for ICLR, but rather the way it is presented is unreadable for the ICLR audience. I used some of the cited papers to understand better the approach. I believe it should be presented more suitably for ML readers, in a self-contained way.
- The theoretical contributions largely build on previous works and are not very involved. Moreover, the associated error if the flow does not follow the Eikonal equations is not characterized or discussed. The *Eikonal equations* are a special case of the *Hamilton-Jacobi Equation*, so focusing on the latter will give more general results.
- The experiments focus on flows that follow the eikonal equation, and it is unclear to me how the approach works otherwise.

# Structure
- Some theorems are given in the related works section



# Dynamical systems terms and difficult to follow
- *flow idealization* -- appears first in the introduction, making it hard to understand the idea for an ML reader
- *robustness to spacial resolution* is unclear in the abstract for people unfamiliar with the Eikonal Equation; perhaps it is better to explain in simple terms that is beneficial
- It is hard to understand how to implement the method proposed in Sec. 3. Provide steps or pseudo code that are self-contained. Explain the symbols in Figure 2 and repeat the most important notation in the caption.


# Writing


**Abstract.**
After reading solely the abstract I could not understand the contributions. As a person not familiar with the Eikonal equation, I was confused. I hope this feedback helps improve it.
-  The motivation posed in the abstract is the lack of approximation rates in terms of depth. The contributions as listed in the abstract are unclear on how they relate to this motivation--what is the rate shown in terms of depth? Please list the precise rate.
- The text in the brackets "(corresponding to time)" is also not very clear. I suggest you avoid writing in brackets here and improve this part.
- The last two sentences are unclear: I did not understand by this point that the goal is to solve the Eikonal equation. I also did not know that the spatial resolution needs to be specified and why we want to be robust to it.



# Minor
- typo line 203: will determined
- use citing with brackets where appropriate

**Questions:**

1. Please elaborate on the motivation to use the Eikonal equations, vs other ones, such as the more general Hamilton-Jacobi Equation, Level Set eq, or other.
2. Elaborate on how the function space considered in your work connects to practical applications. The motivation in the introduction you mention the function spaces considered in Montanelli (2021); Poggio et al. (2017); and He (2023) "lack a clear connection to practical applications". Could you please include a discussion contrasting why the function space in this work relates better to practical applications relative to those in the listed works?
3. What is the optimal rate for the considered setting? Can you contrast that with the optimal rate for the one-dimensional case in Li et al. (2022)?
4. How does your method handle the case when the viscosity solution of an Eikonal equation may not be in the considered target space? Could you provide examples?
5. Does your proposed method work for Hamiltonian Generative Nets?

---

### Official Review · Reviewer_jWqv · 2024-11-08

**Soundness:** 2
**Presentation:** 1
**Contribution:** 2
**Rating:** 3
**Confidence:** 4

**Summary:**

The authors propose a new method for training and learning solutions of the eikonal equation. They also prove an universal approximation theorem that applies (and justifies) the method proposed.

**Strengths:**

From what I could understand, the proposed method is promising.

**Weaknesses:**

- The paper is very hard to read and the writing seems rushed. The paper contains several typos or mathematical inconsistencies, such as:
  - Line 155: Pointing to equation 23 instead of (I think) equation 5.
  - Line 164: Should it not be $\chi_{\epsilon}(\varphi_{V_u}(x,t))$ instead of $\varphi_{\chi_{\epsilon} V_u}(x,t))$?
- There are several definitions missing, such as the definition of flow map in line 155.
- In the approximation Theorem provided by the authors (Theorem 2.1), the definition of the error term is a bit odd, and makes me wonder if a better result could not be achieved. The error term that I am referring to is $E_{\mathcal W}(V_u)=\inf \|W - \tilde V_u\|$. I wonder if it did not make more sense to have $E_{\mathcal{W}}(V_u)=\inf\||x-x_s|W/|W| - \tilde V_u\|$, per the definition of $\tilde V_u$.
- The loss is not explained at all.  $\bar u_\theta$ is not defined.
- The experiments are also insufficient, and there are no comparisons with other PINN methods for solving the eikonal equation.

**Questions:**

- Have the authors considered using Neural ODEs for training the Neural network? I believe it to be a better alternative than the method proposed, at least according to Figure 2.

---

### Note · Authors · 2024-11-15

I have read and agree with the venue's withdrawal policy on behalf of myself and my co-authors.